# Beta-Hydroxyisovaleryl-Shikonin Eradicates Epithelial Cell Adhesion Molecule-Positive Liver Cancer Stem Cells by Suppressing dUTP Pyrophosphatase Expression

**DOI:** 10.3390/ijms242216283

**Published:** 2023-11-14

**Authors:** Yoshiro Asahina, Hajime Takatori, Kouki Nio, Hikari Okada, Takehiro Hayashi, Tomoyuki Hayashi, Tomomi Hashiba, Tsuyoshi Suda, Masaki Nishitani, Saiho Sugimoto, Masao Honda, Shuichi Kaneko, Taro Yamashita

**Affiliations:** Department of Gastroenterology, Graduate School of Medical Science, Kanazawa University, Kanazawa 920-8641, Japan

**Keywords:** hepatocellular carcinoma, dUTP pyrophosphatase, EpCAM, cancer stem cell, Wnt signaling, beta-hydroxyisovaleryl-shikonin

## Abstract

Cancer stem cells (CSCs) play an essential role in tumorigenesis, chemoresistance, and metastasis. Previously, we demonstrated that the development of hepatocellular carcinoma (HCC) is dictated by a subset of epithelial cell adhesion molecule-positive (EpCAM+) liver CSCs with the activation of Wnt signaling. In this study, we evaluated the expression of dUTP pyrophosphatase (dUTPase), which plays a central role in the development of chemoresistance to 5-fluorouracil, in EpCAM+ HCC cells. We further evaluated the effect of beta-hydroxyisovaleryl-shikonin (β-HIVS), an ATP-noncompetitive inhibitor of protein tyrosine kinases, on HCC CSCs. EpCAM and dUTPase were expressed in hepatoblasts in human fetal liver, hepatic progenitors in adult cirrhotic liver, and a subset of HCC cells. Sorted EpCAM+ CSCs from HCC cell lines showed abundant nuclear accumulation of dUTPase compared with EpCAM-negative cells. Furthermore, treatment with the Wnt signaling activator BIO increased EpCAM and dUTPase expression. In contrast, β-HIVS treatment decreased dUTPase expression. β-HIVS treatment decreased the population of EpCAM+ liver CSCs in a dose-dependent manner in vitro and suppressed tumor growth in vivo compared with the control vehicle. Taken together, our data suggest that dUTPase could be a good target to eradicate liver CSCs resistant to 5-fluorouracil. β-HIVS is a small molecule that could decrease dUTPase expression and target EpCAM+ liver CSCs.

## 1. Introduction

Hepatocellular carcinoma (HCC) is one of the most common malignancies and is ranked as the third leading cause of death worldwide [1]. GLOBOCAN data indicate that approximately 840,000 patients were diagnosed with HCC and 780,000 patients died in 2018 [2]. HCC generally develops against a background of chronic liver disease due to hepatitis B and C virus infection, alcohol abuse, and nonalcoholic steatotic liver disease [3]. The survival outcome of HCC patients is stratified using HCC staging systems such as the Barcelona Clinic Liver Cancer stage and the tumor, node, metastasis (TNM) stage. However, the clinical outcome of HCC patients sometimes varies individually, even if patients are diagnosed at the same stage, most likely due to the different malignant natures of the tumors.

The stemness of cancer was historically identified by pathological findings such as poorly differentiated or undifferentiated cellular morphology of tumors, which is generally associated with poor survival outcome after radical treatment [4]. The cancer stem cell (CSC) hypothesis states that a subset of tumor cells with stem-cell-like features play a fundamental role in the maintenance of tumors, and they are considered as pivotal targets to prevent the development of resistance to chemo/radiotherapy in cancer [4,5]. Previously, we demonstrated that epithelial cell adhesion molecule-positive (EpCAM+) HCC cells are CSCs. Isolated EpCAM+ HCC cells are highly tumorigenic and resistant to chemotherapeutic reagents including 5-fluorouracil (5-FU) [6]. EpCAM expression is regulated by Wnt/β-catenin signaling [7], and the genes encoding alpha-fetoprotein (AFP), Dickkopf-1, glypican 3, and EpCAM are coregulated and activated in a subset of HCC cells with stem cell features (hepatic stemlike HCC) [6,7], now widely termed as a progenitor class of HCC [8]. We have shown that EpCAM+ CSCs are resistant to 5-FU, and it is unclear why EpCAM+ cells show chemoresistance to 5-FU.

Previously, by using comprehensive gene expression analysis, we found that a subset of HCC cells with activation of the *DUT* gene, encoding dUTP pyrophosphatase (dUTPase), correlate with poor prognosis after surgery [9]. Furthermore, *DUT* knockdown results in the sensitization of HCC cells to 5-FU, suggesting that dUTPase expression is associated with the biological malignant nature and 5-FU resistance of HCC [9]. However, the relationship between EpCAM and dUTPase expression as well as 5-FU resistance in EpCAM+ CSCs remains unclarified.

In the present study, we evaluated dUTPase expression in EpCAM+ HCC cells. We further evaluated the effect of beta-hydroxyisovaleryl-shikonin (β-HIVS), which is an ATP-noncompetitive inhibitor of protein tyrosine kinases that also suppresses dUTPase expression [10], on HCC.

## 2. Results

### 2.1. EpCAM and dUTPase Are Expressed in Fetal Hepatoblasts, Adult Biliary Epithelial/Progenitor Cells, and a Subset of HCC Cells

We examined the expression of EpCAM and dUTPase in human fetal liver (10 weeks) and human adult normal liver by immunohistochemistry (IHC) (Figure 1). In fetal liver, the membranous expression of EpCAM (Figure 1, left upper panel) and nuclear/cytoplasmic expression of dUTPase (Figure 1, left lower panel) were detected in hepatoblasts. In contrast, EpCAM and dUTPase were not expressed in hepatocytes but were detected in biliary epithelial cells (Figure 1, black arrows in right upper and lower panels).

In cirrhotic liver, EpCAM and dUTPase expression was detected by double-color IHC analysis in ductular reactions, where liver progenitor cells are supposed to be located (Figure 2, left panel). EpCAM and dUTPase expression was also confirmed in a subset of HCC cells with portal vein invasion (middle and right panels in Figure 2). These data suggest the coexpression of dUTPase and the hepatic stem/biliary marker EpCAM in fetal and adult liver tissues and HCC tissues.

### 2.2. dUTPase as a Marker of Poor Prognosis in HCC Patients

Next, the prognostic outcomes of 107 HCC patients with high or low dUTPase expression who received surgery were evaluated. The clinicopathological characteristics of the HCC patients are shown in Table 1. IHC scores were used to classify HCC with high/low dUTPase expression according to our previous study [9], and 68 and 39 patients with HCC were regarded as dUTPase-low or -high, respectively.

Kaplan–Meier survival analysis showed that dUTPase-high HCC patients had poorer overall survival (*p* = 0.009) and recurrence-free survival (*p* = 0.0016) than dUTPase-low HCC patients (Figure 3). dUTPase-high HCC was associated with poorly differentiated morphology (*p* = 0.0002, chi-square test). dUTPase-high HCC was also associated with high serum AFP values (≥100 ng/mL) with borderline significance (*p* = 0.079, chi-square test). Noticeably, dUTPase-high HCC patients showed frequent expression of EpCAM compared with dUTPase-low HCC patients (*p* = 0.012) (Table 2). Univariate and multivariate Cox regression analyses showed that high dUTPase expression is an independent prognostic factor involved in HCC death (Table 3). Taken together, our data suggest that dUTPase expression is associated with poorly differentiated morphology accompanied by an elevation of the hepatic progenitor markers EpCAM and AFP, and high dUTPase expression correlates with short survival after surgery due to the high incidence of tumor recurrence.

### 2.3. Regulation of dUTPase Expression by Wnt/β-Catenin Signaling

Wnt/β-catenin signaling is an important regulator of CSC maintenance. Our data suggest that EpCAM and dUTPase expression overlaps to some extent in hepatoblasts, biliary epithelial cells, and liver CSCs. Since we previously demonstrated that EpCAM expression is regulated by Wnt/β-catenin signaling, the effect of Wnt/β-catenin signaling modulation on dUTPase expression was examined. Huh7 cells were transfected with the plasmids pGL3-DUT-full, which reflects the promoter activity of the gene encoding dUTPase (*DUT*), and pRL-SV40 as a control. These cells were treated with BIO, which activates Wnt/β-catenin signaling by inhibiting GSK 3β to stabilize β-catenin, or control Me-BIO, which has no effect on GSK 3β [6]. Compared with MeBIO-treated cells, BIO-treated cells exhibited a twofold increase in *DUT* promoter activity (Figure 4, left panel). These transfected cells were further treated with small interfering RNAs (siRNAs; si-1 CTNNB1 and si-2 CTNNB1) targeting *CTNNB1*, which encodes β-catenin. As a result, dUTPase gene promoter activity was reduced by *CTNNB1* knockdown (Figure 4, right panel). These data suggest that dUTPase gene expression is regulated by Wnt/β-catenin signaling.

Next, the EpCAM+ and EpCAM-negative (EpCAM−) cell populations were sorted by fluorescence-activated cell sorting (FACS), and the status of dUTPase and β-catenin was evaluated by immunofluorescence. Sorted EpCAM+ cells clearly showed the nuclear accumulation of dUTPase and β-catenin compared with EpCAM− cells (Figure 5).

Double-color IHC analysis was then used to evaluate EpCAM, dUTPase, and β-catenin expression in human liver cirrhosis and HCC tissues (Figure 6). In cirrhotic liver, EpCAM, β-catenin, and dUTPase were clearly detected and overlapped in ductular reactions (Figure 6, red arrows in left panels). Similarly, EpCAM, β-catenin, and dUTPase were expressed in a subset of HCC cells (Figure 6, yellow arrows in right panels). Taken together, these data suggest that dUTPase is overexpressed in EpCAM+ liver progenitors and CSCs potentially through the activation of Wnt/β-catenin signaling.

### 2.4. β-HIVS Suppresses dUTPase Expression and EpCAM+ Liver CSCs to Inhibit HCC Growth

Since dUTPase plays an important role in the DNA damage response by inhibiting uracil misincorporation [11,12], candidate chemicals that could suppress dUTPase expression or activity were searched for in a literature-based manner. As a result, one study was found indicating that β-HIVS suppresses dUTPase expression in vitro [10]. β-HIVS is isolated from the roots of the traditional oriental medicinal herb *Lithospermum radix* [10]. A previous report indicated that β-HIVS could suppress several lung cancer cell lines by inducing apoptosis [13].

Accordingly, Huh7 cells were treated with β-HIVS for 48 h, which resulted in a reduction in dUTPase expression (Figure 7a) and the EpCAM+ liver CSC population (Figure 7b) in a dose-dependent manner. The effect of β-HIVS on HCC growth was evaluated further using Huh1 cells in vivo. A Two-week treatment with β-HIVS suppressed HCC growth compared with the control (*p* = 0.036) (Figure 7c,d). Taken together, these data suggest the utility of β-HIVS for the treatment of progenitor-type HCC in reducing the EpCAM+ CSC population by suppressing dUTPase expression.

## 3. Discussion

Carcinogenesis and embryogenesis share common features in terms of cell proliferation and motility, interaction with the stroma, and existence of stemlike cells [4]. Molecularly, HCC can be classified into several types with distinct phenotypes and signaling pathway activation [14]. Previously, we demonstrated that aggressive HCC can be classified according to certain developmental stages of the liver based on the expression status of the stem cell markers EpCAM and AFP [6]. This HCC class is termed progenitor-type HCC according to the consensus molecular classification of HCC [8]. We demonstrated that progenitor-type HCC is resistant to 5-FU, potentially due to the presence of chemoresistant EpCAM+ liver CSCs [6]. In the present study, we showed that one of the key nucleotide-metabolizing enzymes, dUTPase, which plays a critical role in the development of 5-FU resistance [12], was activated in EpCAM+ liver CSCs. Furthermore, we revealed that chemical inactivation of dUTPase by β-HIVS could eradicate EpCAM+ liver CSCs and suppress tumor growth in vivo. The findings of this study suggest the utility of targeting dUTPase to treat progenitor-type HCC with poor prognosis.

Although 5-FU is one of the most widely used chemotherapeutic reagents for the treatment of various types of cancer, it is not used for the treatment of HCC except in combination with cisplatin as hepatic arterial infusion chemotherapy in some regions. One of the major reasons for this is 5-FU resistance in HCC, and it is an important task to elucidate the mechanism of 5FU resistance. Thymidylate synthase is the key enzyme required for DNA synthesis and is targeted by 5-FU. Long-term 5-FU treatment in cancer cells results in the acquisition of resistance to 5-FU, and the upregulation of thymidylate synthase is one of the mechanisms attenuating the effect of 5-FU to reduce thymidylate depletion and concurrent DNA damage. The other mechanism preventing the DNA damage induced by 5-FU treatment is the upregulation of dUTPase. dUTPase converts dUTP to dUMP and prevents the accumulation of dUTP, which can be integrated into DNA by errors to induce the DNA damage response. Thus, dUTPase activation could reduce the DNA damage response caused by uracil misincorporation, which occurs naturally by nucleotide metabolism or by 5-FU treatment [12]. Human dUTPase is encoded by the DUT gene, and overexpression of DUT was found in 42% of HCC tumors. DUT is also involved in sorafenib resistance via activation of the NF-κB transcription factor [15].

We found that dUTPase was abundant in fetal hepatoblasts, adult liver progenitors, and liver CSCs and was potentially regulated by Wnt/β-catenin signaling. Wnt/β-catenin signaling is activated during embryogenesis and normal liver development [16]. During embryogenesis, cells divide to generate normal organs, and active cell division may result in the activation of nucleotide metabolism and the accumulation of dUTP misincorporation, which might cause replication errors or genetic mutations in daughter cells. Our data suggest that dUTPase is activated by Wnt/β-catenin signaling, and one potential benefit obtained by dUTPase activation might be the reduction in DNA damage to correctly transfer genome information to daughter cells during embryogenesis with Wnt/β-catenin signaling activation. Genomic sequence information revealed a putative binding element to TCF, a promoter of Wnt/β-catenin signaling, in the enhancer region of the *DUT* gene (Appendix A). Although further studies are needed, a mechanism of dUTPase activation by Wnt/β-catenin signaling is suspected. Liver CSCs might hack the same system and reduce the accumulation of DNA damage, which is closely related to 5-FU resistance.

We demonstrated that β-HIVS could reduce dUTPase expression and the population of EpCAM+ liver CSCs and inhibit HCC growth. β-HIVS is isolated from the roots of the traditional oriental medicinal herb *Lithospermum radix* [10]. Several studies reported the mechanisms of β-HIVS regarding its inhibitory effect on tumor growth. A previous report indicated that β-HIVS could suppress several lung cancer cell lines by inducing apoptosis [13]. One point we should note is that β-HIVS might also exert anticancer effects through the inhibition of tyrosine kinases such as v-Src, EGFR, or VEGFR2 [10,17]. PI3K/AKT signaling has also been reported as a signal involved in the inhibitory effect on tumor growth of β-HIVS [18,19]. Thus, future studies are required to investigate if the effect of β-HIVS on tumor growth inhibition with reduced EpCAM expression depends on the reduction in dUTPase alone. Recently, TAS-114, a potent inhibitor of dUTPase to enhance the antitumor activity of 5-FU, was clinically tested in lung and gastric cancer patients [20,21]. In HCC, TAS114 could potentiate suppression of HCC growth that synergized with sorafenib [22]. dUTPase inhibition might be a good approach in targeting liver CSCs for the treatment of progenitor-type HCC with poor prognosis. Moreover, dUTPase inhibitors are also expected to overcome the chemo/radio-resistance associated with elevated dUTPase in multiple bridging therapies, including TACE and radioembolization for HCC; they have potential applications in sequential therapy for posttreatment recurrence and in combination with existing therapies.

## 4. Materials and Methods

### 4.1. Patients

A total of 107 patients were diagnosed with HCC and received surgery at Kanazawa University Hospital. Surgically resected HCC samples were used to examine EpCAM and dUTPase expression, as described previously [9]. The clinical information of the patients was collected retrospectively from medical records. This study conformed to the standards set by the Declaration of Helsinki, and the protocol was approved by the Institutional Review Board of the Graduate School of Medical Sciences, Kanazawa University (IRB no.: 2016-093). Patients provided written informed consent.

### 4.2. Cell Lines and Reagents

The Huh7 and Huh1 HCC cell lines previously identified as positive for EpCAM and CD133 [6] were obtained from the Japanese Collection of Research Bioresources Cell Bank (Osaka, Japan). The cells were maintained at 37 °C in Dulbecco’s modified Eagle’s medium (Gibco, Grand Island, NY, USA) supplemented with 10% fetal bovine serum (Gibco).

### 4.3. Immunohistochemistry

Formalin-fixed, paraffin-embedded tissues were prepared for IHC staining. Briefly, the slides were deparaffinized and rehydrated, antigen retrieval was performed by citrate buffer, and slides were blocked by Protein Block Serum-Free (Dako, Carpinteria, CA, USA). The slides were incubated with the following primary antibodies overnight at 4 °C: anti-dUTPase (M01; Abnova Corporation, Taipei, Taiwan), anti-EpCAM (VU-1D9; Oncogene Research Products, San Diego, CA, USA), and anti-β-catenin (clone 14; BD Biosciences, San Jose, CA, USA). The slides were processed using an Envision+ Kit (Dako). Vector Blue and Red Kits (Vector Laboratories, Inc., Burlingame, CA, USA) were used for double-color IHC. IHC images were analyzed as described previously [9]. Briefly, the staining area was evaluated and scored on four levels (none = 0, focal = 1, multifocal = 2, and diffuse = 3). Staining intensity was evaluated and scored on three levels (none = 0, mild = 1, and strong = 2), and the sum of both levels was used as the immunostaining score (0–5). dUTPase expression in the tumor was defined as low (≤2 points) or high (≥3 points).

### 4.4. FACS Analysis

Huh7 cells were trypsinized, washed, and resuspended in Hank’s balanced salt solution (Lonza, Basel, Switzerland) supplemented with 1% HEPES and 2% phosphate-buffered saline. The cells were incubated with a fluorescein isothiocyanate-conjugated or allophycocyanin-conjugated anti-EpCAM monoclonal antibody (BER-EP4; Dako) on ice for 30 min prior to cell sorting using a FACSAria II (BD Biosciences). Sorted cells were harvested on dishes and cultured overnight for image analysis.

### 4.5. Western Blot Analysis

Whole cell lysates were prepared from cells using RIPA lysis buffer. The primary antibodies used for Western blot analysis were an anti-dUTPase monoclonal antibody (M01 clone 1C9; Abnova Corporation) and anti–β-actin antibody (Cell Signaling Technology, Inc., Danvers, MA, USA). Immune complexes were visualized using enhanced chemiluminescence detection reagents (Amersham Biosciences Corp., Piscataway, NJ, USA).

### 4.6. RNA Interference

siRNAs targeting human β-catenin (s437 and s438; Thermo Fisher Scientific, Waltham, MA, USA) and a nonspecific control (Silencer Select Negative Control No. 1; Applied Biosystems, Foster City, CA, USA) were purchased. Cells grown to 60–80% confluency in 6-well plates were transfected with siRNAs (100 pmol/L) using Lipofectamine^®^ 2000 (Invitrogen Life Technologies, Carlsbad, CA, USA).

### 4.7. Immunofluorescence

Cells were fixed in cold 100% methanol and blocked with Protein Block Serum-Free (Dako). Anti-dUTPase antibody (M01; Abnova Corporation) and anti-β-catenin antibody (clone 14; BD Biosciences) were used as primary antibodies. Alexa 488 fluorescein isothiocyanate-conjugated anti-mouse IgG (Thermo Fisher Scientific) was used as a secondary antibody. Confocal fluorescence microscopic analysis was performed essentially as described previously [23].

### 4.8. Luciferase Reporter Assay

A plasmid including the *DUT* promoter (pGL3-DUT-full) was kindly provided by Dr. Robert D. Ladner [24]. pRL-SV40 was cotransfected as an internal control. Huh7 cells grown to 60–80% confluency in 6-well plates were transfected with 0.5 μg of each plasmid using Lipofectamine^®^ 2000 (Invitrogen Life Technologies). Luciferase activity was measured in triplicate with the Dual-Luciferase Reporter Assay System and GloMax-Multi Detection System (Promega, Madison, WI, USA).

### 4.9. Animal Studies

NOD.CB17-Prkdcscid/J (NOD/SCID) male mice were purchased from Charles River Laboratories, Inc. (Wilmington, MA, USA). The mice were housed under specific pathogen-free conditions with a 12 h light/dark cycle and provided with tap water and food ad libitum. Huh7 cells (approximately 1.0 × 10^6^ cells) were resuspended in 200 μL of a 1:1 Dulbecco’s modified Eagle’s medium:Matrigel (BD Biosciences) mixture. β-HIVS (Wako Pure Chemical Industries, Osaka, Japan) was solved in saline containing 5% dimethyl sulfoxide, 7.5% Cremophor EL, 7.5% ethanol, and 5% glucose. Control vehicle (200 μL/body) or β-HIVS (30 mg/kg/day, 200 μL/body) was injected intraperitonially three times/week for 2 weeks. Subcutaneous tumor volume was recorded as described previously [14]. The experimental protocol was approved by the Kanazawa University Animal Care and Use Committee and conformed to the Guide for the Care and Use of Laboratory Animals prepared by the National Academy of Sciences.

### 4.10. Statistical Analysis

Chi-square tests, *t*-tests, Mann–Whitney’s *U* tests, and Kaplan–Meier survival analyses with log-rank tests were performed using Prism 7 (GraphPad Software, San Diego, CA, USA). Univariate and multivariate Cox regression analyses were performed for identifying the factors associated with patient prognosis using SPSS software (ver. 28.0; IBM Japan, Ltd., Tokyo, Japan). Statistical significance was set at *p* < 0.05.

## Figures and Tables

**Figure 1 ijms-24-16283-f001:**
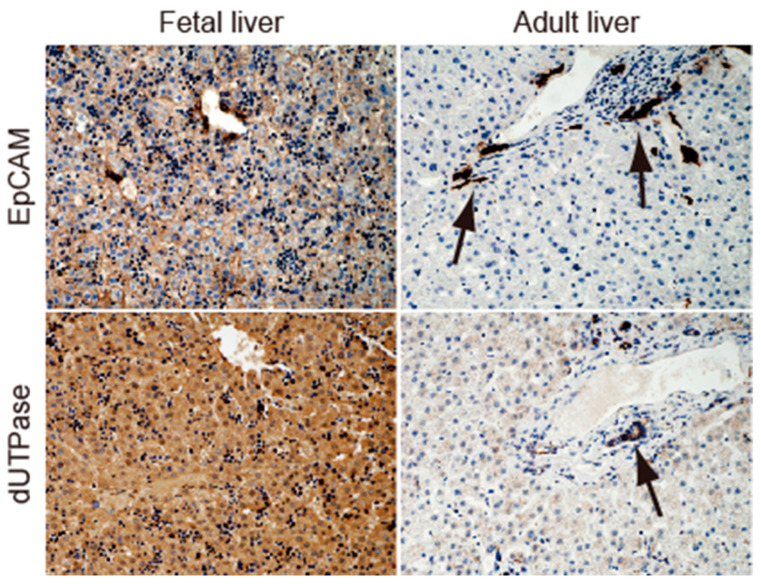
EpCAM and dUTPase expression in normal human fetal (**left panels**) and adult (**right panels**) liver tissues. Black arrows indicate bile duct epithelial cells in adult liver.

**Figure 2 ijms-24-16283-f002:**
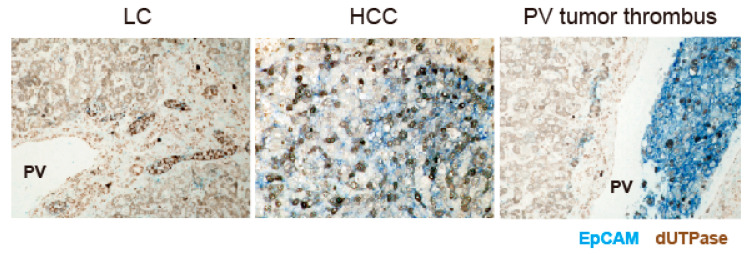
EpCAM (stained with blue dye) and dUTPase (stained brown with 3,3′-diaminobenzidine) expression in human liver cirrhosis tissue (left panel) and HCC tissue (middle and right panels) by double-color IHC. Weak membranous EpCAM and strong cytoplasmic dUTPase expression was detected in ductular reactions (**left panel**). Modest membranous EpCAM and nuclear dUTPase expression was observed in HCC (**middle panel**). High levels of membranous/cytoplasmic EpCAM expression and nuclear dUTPase expression were detected in a tumor thrombus within the portal vein (PV) (**right panel**).

**Figure 3 ijms-24-16283-f003:**
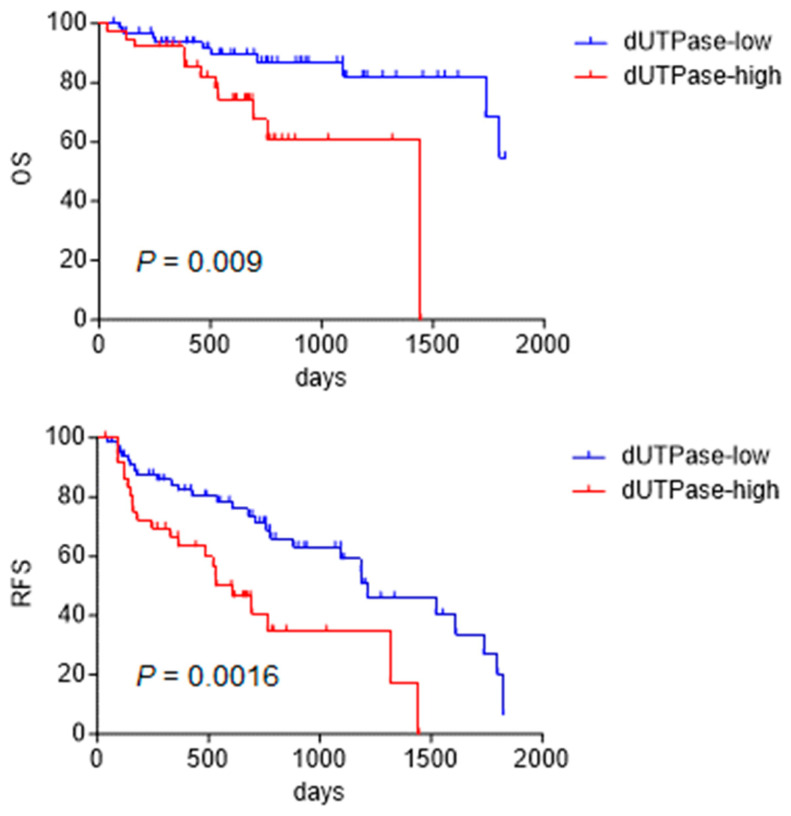
Kaplan–Meier survival analysis of HCC patients (*n* = 107) who received surgery according to dUTPase expression status. dUTPase-high HCC patients showed significantly worse overall survival (OS, **upper panel**) and recurrence-free survival (RFS, **lower panel**) compared with dUTPase-low HCC patients.

**Figure 4 ijms-24-16283-f004:**
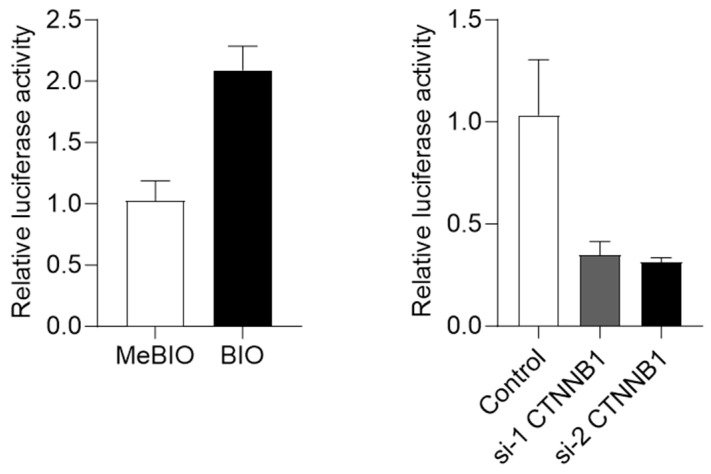
Regulation of *DUT* (encoding dUTPase) promoter activity by Wnt/β-catenin signaling. Compared with Me-BIO, BIO treatment activated *DUT* promoter activity (**left panel**). In contrast, inactivation of Wnt/β-catenin signaling by *CTNNB1* knockdown inhibited *DUT* promoter activity (**right panel**).

**Figure 5 ijms-24-16283-f005:**
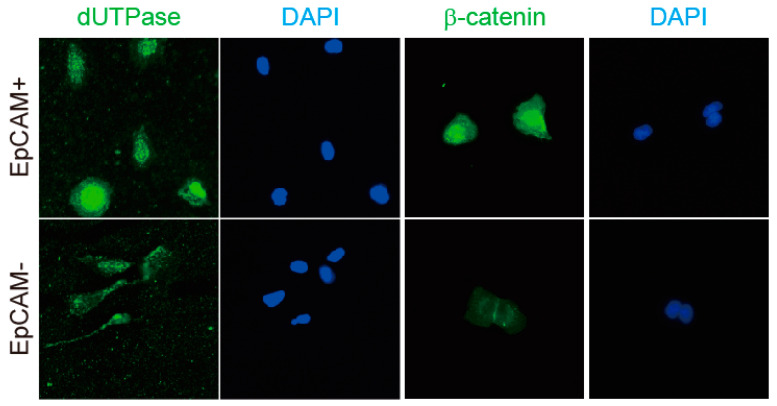
dUTPase and β-catenin expression in sorted EpCAM+ and EpCAM− cells. Sorted EpCAM+ cells clearly showed nuclear dUTPase and β-catenin expression compared with EpCAM− cells.

**Figure 6 ijms-24-16283-f006:**
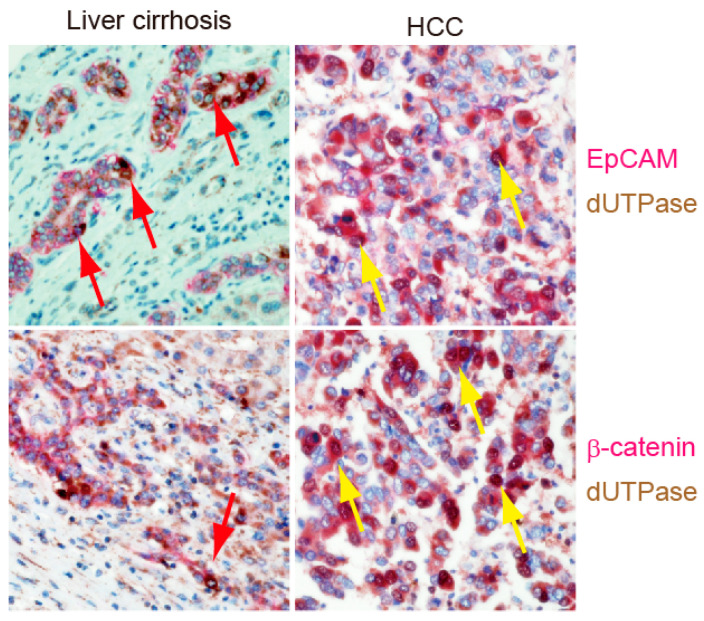
EpCAM (stained with red dye, **upper panels**), β-catenin (stained with red dye, **lower panels**), and dUTPase (stained brown with 3,3′-diaminobenzidine, all panels) expression in human liver cirrhosis tissue (**left panels**) and HCC tissue (**right panels**) by double-color IHC. Red and yellow arrows indicate the presence of normal liver progenitor cells and liver CSCs, respectively.

**Figure 7 ijms-24-16283-f007:**
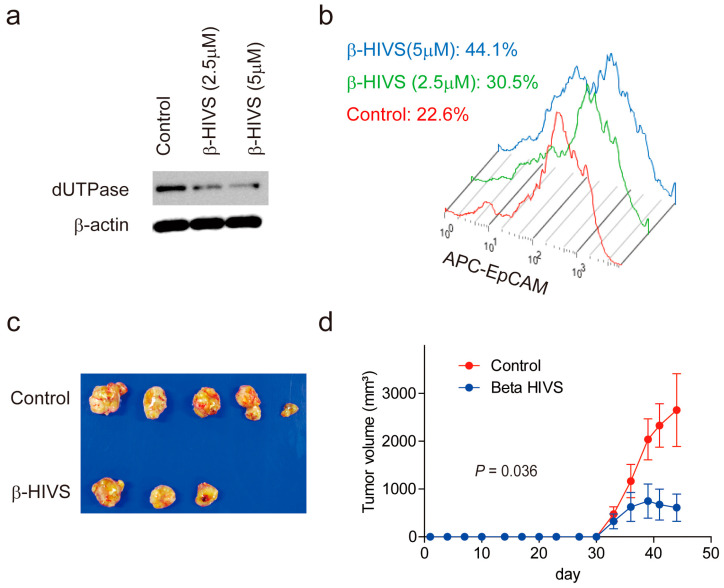
Inhibition of HCC growth by β-HIVS. (**a**) Western blot images of cell lysates obtained from Huh7 cells treated with control dimethyl sulfoxide or β-HIVS at the indicated concentration for 48 h. (**b**) FACS analysis of Huh7 cells treated with control dimethyl sulfoxide or β-HIVS at the indicated concentration (0 μM, red line; 2.5 μM, green line; 5 μM, blue line) for 48 h. The cells were labeled with allophycocyanin (APC)-conjugated anti-EpCAM antibodies. The percentages of the EpCAM− population are indicated. (**c**) A photomicrograph of subcutaneous tumors originating from Huh1 cells and developed in NOD/SCID mice treated with control vehicle or β-HIVS for 2 weeks. (**d**) Tumor volume curves of subcutaneous tumors originating from Huh1 cells and developed in NOD/SCID mice treated with control vehicle (red line) or β-HIVS (blue line) for 2 weeks.

**Table 1 ijms-24-16283-t001:** Characteristics of HCC patients.

	Patients with HCC (*n* = 107)
Age, years (mean, minimum–maximum)	64, 39–80
Sex, male/female	82/25
Etiology, HBV/HCV/HBV + HCV/others	33/65/3/6
LC, yes/no	69/38
UICC (8th) TNM stage, I–II/III–IV	79/28
Histological grade, well/moderate/poor	19/75/13
Tumor size ≥3 cm/<3 cm	42/65
Serum AFP, median, 25–75% percentile (ng/mL)	15, 10.0–133.0

**Table 2 ijms-24-16283-t002:** Characteristics of HCC patients according to dUTPase status.

	dUTPase-Low (*n* = 68)	dUTPase-High (*n* = 39)	*p*
Age (mean)	64	64	0.86
Sex, male/female	50/18	32/7	0.35
Etiology, HBV/HCV/HBV + HCV/others	19/42/2/5	14/23/1/1	0.67
LC, yes/no	43/25	26/13	0.83
UICC (8th) TNM stage, I–II/III–IV	52/16	27/12	0.49
Histological grade, well/moderate/poor	16/50/2	3/25/11	0.0002
Tumor size ≥3 cm/<3 cm	27/41	15/24	1.0
Serum AFP ≥100/<100 (ng/mL)	16/52	16/23	0.079
EpCAM expression (positive/negative)	18/50	20/19	0.012

**Table 3 ijms-24-16283-t003:** Univariate and multivariate Cox regression analyses of HCC death.

	Univariate		Multivariate	
Variables	HR (95% CI)	*p* Value	HR (95% CI)	*p* Value
Age (years; ≥60/<60)	0.88 (0.38–2.0)	0.76		
Sex (female/male)	0.86 (0.32–2.3)	0.76		
Etiology (viral/non-viral)	1.42 (0.19–10.6)	0.74		
LC (yes/no)	1.83 (0.67–5.0)	0.24		
UICC (8th) TNM stage (III–IV/I–II)	2.85 (1.2–6.8)	0.018	2.63 (1.08–6.4)	0.033
Histological grade (moderate or poor/well)	2.86 (0.78–10.5)	0.11		
Tumor size (≥3 cm/<3 cm)	1.35 (0.56–3.25)	0.5		
Serum AFP (≥100/<100 ng/mL)	2.09 (0.88–4.97)	0.097	N.E.	
dUTPase (high/low)	3.19 (1.31–7.75)	0.011	3.03 (1.23–7.51)	0.016

## Data Availability

Data are available from the corresponding authors upon reasonable request.

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
