# Peer review of "Beta-Hydroxyisovaleryl-Shikonin Eradicates Epithelial Cell Adhesion Molecule-Positive Liver Cancer Stem Cells by Suppressing dUTP Pyrophosphatase Expression"

_ijms, 2023, doi:10.3390/ijms242216283_

Round 1

Reviewer 1 Report

Comments and Suggestions for Authors

This study investigates the role of dUTPase in HCC stem cells, especially in the context of 5-FU resistance. However, several deficiencies were identified, as follow:

1. The whole premise of HCC stem cells being resistant to 5-FU should be based on its clinical relevance. However, 5-FU is not FDA-approved for HCC treatment, nor it is a first-line therapy in HCC in most countries. This must be thoroughly justified why the authors chose this drug. Nevertheless, this concern diminishes translational value of authors’ findings.

2. Figure 2. The use of EPCAM as a single marker for identifying HCC stem cells is insufficient. At least one more HCC stemness marker, e.g. CD133, should be evaluated in parallel.

3. Figure 3. Please perform multivariate Cox regression analysis to evaluate whether dUTPase is an independent marker for patient prognosis. In any case, a similar result has already been reported in PMID: 19968781.

4. Figure 4. It is unclear whether CTNNB1 or its associated TFs directly binds to the dUTPase promoter. Please validate such interaction with ChIP-PCR, and also supplement these data with qPCR and western blot data after si-CTNNB1 treatment. 

5. It is unclear whether the action of β-HIVS is actually dependent on the down-regulation of dUTPase. One should assess if β-HIVS has differential effects on EPCAM+ and EPCAM- cells, and perform drug treatment experiments with dUTPase overexpression or KO.

Comments on the Quality of English Language

The quality of English is fine.

Reviewer 2 Report

Comments and Suggestions for Authors

Good paper.

1. You have not added any recent article while discussing drug resistance.

Xu M, Liu Y, Wan HL, Wong AM, Ding X, You W, Lo WS, Ng KK, Wong N. Overexpression of nucleotide metabolic enzyme DUT in hepatocellular carcinoma potentiates a therapeutic opportunity through targeting its dUTPase activity. Cancer Lett. 2022 Nov 1;548:215898. doi: 10.1016/j.canlet.2022.215898. Epub 2022 Sep 6. PMID: 36075487.

2. Can you please add a paragraph regarding suppression of dUTPase in the patients treated with drugs like soragenib (first line) and other like regorafenib which has recently been approved for second line treatment.

3. Can you please add few lines on the effect of multiple bridging therapy like TACE, radioembolization on the dUTPase expression. 

Author Response

Dear Reviewer,

We thank the reviewers for providing the constructive criticism which has helped us to improve our manuscript. We have revised the text in accordance with the reviewers’ comments.

We hope that the revised manuscript is now suitable for publication. 

  1. You have not added any recent article while discussing drug resistance.

Xu M, Liu Y, Wan HL, Wong AM, Ding X, You W, Lo WS, Ng KK, Wong N. Overexpression of nucleotide metabolic enzyme DUT in hepatocellular carcinoma potentiates a therapeutic opportunity through targeting its dUTPase activity. Cancer Lett. 2022 Nov 1;548:215898. doi: 10.1016/j.canlet.2022.215898. Epub 2022 Sep 6. PMID: 36075487.

 > We greatly appreciate the reviewer’s suggestion. We have cited the relevant article and added its content.

  1. Can you please add a paragraph regarding suppression of dUTPase in the patients treated with drugs like soragenib (first line) and other like regorafenib which has recently been approved for second line treatment.

> We appreciate the reviewer's comments. Unfortunately, there are no data on dUTPase suppression in patients treated with TKIs such as sorafenib or regorafenib. Since the authors of article above showed that DUT is involved in sorafenib resistance via activation of the NF-κB transcription factor(8), we speculate that there is a clinical correlation between high expression of dUTPase and resistance to TKIs. Future studies are needed to assess the clinical correlation. In addition, from the relevant article, TAS-114, a potent inhibitor of dUTPase, appears to be promising not only for overcoming 5FU resistance but also for combination therapy with sorafenib. These details have been added to the discussion.

  1. Can you please add few lines on the effect of multiple bridging therapy like TACE, radioembolization on the dUTPase expression. 

> We appreciate the reviewer's comments. Although this study did not identify actual dUTPase elevation in clinical TACE cases, as the reviewer noted, chemo/radio-resistance associated with elevated dUTPase expression in tumor tissue should be considered in the treatment course of HCC with multiple bridging therapies, including TACE and radioembolization. Since dUTPase inhibitors are expected to overcome the chemo/radio-resistance associated with elevated dUTPase in multiple bridging therapies for HCC, they have potential applications in sequential therapy for posttreatment recurrence and in combination with existing therapies. We have added these statements to the last few lines of discussion.

Round 2

Reviewer 1 Report

Comments and Suggestions for Authors

The paper is suitably improved.